# Realworld 3D Object Recognition Using a 3D Extension of the HOG Descriptor and a Depth Camera

**DOI:** 10.3390/s21030910

**Published:** 2021-01-29

**Authors:** Cristian Vilar, Silvia Krug, Mattias O’Nils

**Affiliations:** 1Department of Electronics Design, Mid Sweden University, Holmgatan 10, 851 70 Sundsvall, Sweden; mattias.onils@miun.se (M.O.); silvia.krug@imms.de (S.K.); 2System Design Department, IMMS Institut für Mikroelektronik- und Mechatronik-Systeme Gemeinnützige GmbH (IMMS GmbH), Ehrenbergstraße 27, 98693 Ilmenau, Germany

**Keywords:** 3D object recognition, 3DHOG, histogram-of-gradients, ModelNet40, ModelNet10, feature descriptor, Intel RealSense, depth camera, PCA

## Abstract

3D object recognition is an generic task in robotics and autonomous vehicles. In this paper, we propose a 3D object recognition approach using a 3D extension of the histogram-of-gradients object descriptor with data captured with a depth camera. The presented method makes use of synthetic objects for training the object classifier, and classify real objects captured by the depth camera. The preprocessing methods include operations to achieve rotational invariance as well as to maximize the recognition accuracy while reducing the feature dimensionality at the same time. By studying different preprocessing options, we show challenges that need to be addressed when moving from synthetic to real data. The recognition performance was evaluated with a real dataset captured by a depth camera and the results show a maximum recognition accuracy of 81.5%.

## 1. Introduction

Since the popularization of consumer depth cameras, 3D data acquisition and surface description techniques enabled a significant number of new applications like object recognition [1], robot grasping [2], 3D reconstruction [3] or autonomous navigation [4]. In addition to the visual information, depth cameras allow to detect and measure 3D features regarding the object’s shape, improving therefore the recognition performance in comparison with visual cameras. Additionally, depth cameras are not affected by illumination changes and therefore are especially suitable for safety applications.

Advanced object recognition approaches are typically based on convolutional neural networks (CNNs) to extract a hierarchy set of abstract features from each object to capture key information regarding the class to which it belongs. Despite their recognition performance, CNN approaches demand high computational and memory resources, making them difficult to implement when there are strong limitations in hardware [5]. In addition, CNNs require extended datasets for training and testing which, in practice, limit their application [6].

In opposite to CNNs, classic object recognition approaches rely on a previous expertise to extract a set of key object features to construct a hand-crafted object descriptor. Our previous research on hand-crafted object descriptors showed good object recognition performances in comparison with the state-of-the-art 3D CNN approaches based on the synthetic ModelNet10 dataSet [7] as well as a relatively low processing time making it suitable for real-time processing [8]. The goal of that paper was to develop a processing pipeline using a more traditional approach while maintaining a comparable recognition accuracy to CNN-based approaches. We used the synthetic ModelNet10 dataset for this purpose because recognition results of various other works are available and thus allow a comparison without reimplementing the approaches. However, a synthetic dataset does not allow to evaluate the object descriptor for a real object recognition application using a depth camera which is the intent of this research. Depth cameras provide a non-ideal object measurements with respect the ideal object shapes of the existing synthetic dataset. These non-idealities are caused mainly by noise and depth computation artifacts as well as by the impossibility to capture a complete 3D object shape due to occlusion effects.

Ideally, any 3D object recognition approach targeting real data would be trained on a corresponding real dataset using the 3D objects captured by the depth camera as well as test the recognition processing chain. This involves to measure a relevantly large number of different objects belonging to each class which, in practice, limits the application scope due the impossibility, in many cases, to measure enough objects. This problem is especially significant for the CNN approaches due to the required extended and labeled datasets [6].

The scientific contribution of this work is to use our previously developed processing pipeline as base to explore a 3D object recognition approach using real data from a depth camera for classification and a synthetic dataset to train the classifier. We analyze whether the constraint of large datasets with real data can be leveraged by using synthetic data for the training process. In addition, we systematically review the required data processing steps to maximize the recognition performances and thus extend our previous pipeline to handle real data. To evaluate the depth camera non-idealities in the recognition processing chain, we propose a 3D object recognition approach using a feature-based object descriptor and compare its recognition performance for various combinations of test and training data as well as different preprocessing steps in the classification data flow.

## 2. Related Works

3D object recognition is a fundamental research topic in image processing. Most of the recognition approaches are still based on 2D visual images and thus do not fully exploit the shape information of the objects. Depth cameras allow to measure 3D information of the scene and thus enable 3D perception of the objects. However, 3D measurements result in unstructured point cloud data and are thus highly unorganized, requiring more complex and computational demanding approaches [9].

3D recognition approaches can be categorized into deep-learning or feature descriptor methods. Novel deep-learning approaches are based on 3D CNNs to extract abstract features from the objects regarding their belong class. There are several CNN approaches for 3D recognition, like PointNet [10], VoxNet [11], 3DYolo [12] or SPNet [13]. However, these approaches are highly computational and memory demanding. In addition, they require extended and structured training datasets, making it difficult to implement them in a real recognition application with small datasets, hardware and memory resources.

Feature descriptor approaches, instead, rely on previous expertise to extract the key set of object features to maximize the class separability. The approaches can categorized in methods using global features and local features. Local features describe the local geometry around the key points of interest. Thus, they require the previous selection of valid key points and specialized classifiers to handle multiple feature vectors per object [6]. However, they are robust to clutter and occlusions and thus can be used for 3D object recognition in cluttered scenes. There is a large variety of approaches based on local features, such as scale-invariant feature transform (SIFT) [14], signature of geometric centroids (SGCs) [15], signature of histograms of orientations (SHOTS) [16] or rotational contour signature (RCS) [17]. Local key point detection requires a detailed object resolution in order to extract the key points, leading to poor descriptiveness and computational demanding approaches [18]. As a result, the mentioned descriptors show weak results when using consumer depth cameras due to their image resolution limitations [19].

Global features, instead, encode the whole object shape information as a single feature vector. They require previous object segmentation as a processing step in order to isolate the object and compute the feature descriptor [20], but they are efficient in terms of computational cost and memory consumption [21], achieving good results and popularity [9]. There is an extensive set of approaches based on global 3D features, like spin images (SI) [22], view point feature histogram (VFH) [23], fast point feature histogram (FPFH) [24], ensemble of shape functions (ESF) [25], the novel voxelized fractal descriptor (VFD) [6], the globally aligned spatial distribution (GASD) [19] or the triple local coordinate images (TriLCI) [18]. Another popular 2D-based global descriptor is the histogram-of-oriented-gradients (HOG) [26] which has been extended to 3D (3DHOG) [27,28,29]. However, 3DHOG tends to generate higher dimensionality features and thus requires to include dimensionality reduction methods [28,30]. In addition, existing 3DHOG evaluations use synthetic dataset to compare the descriptor performances [30].

None of the described approaches have so far been evaluated regarding a real application using 3D camera data, which is the intention of this study. Our previous results using a 3DHOG descriptor-based processing pipeline are compared with other methods in Vilar et al. [8] using the ModelNet10 synthetic dataset as a common reference. This showed, that our approach can achieve comparable results while limiting the processing effort and duration. In this publication, we extend the previously tested pipeline in order to handle both synthetic and real data and evaluate if such a combination can leverage the need for extensive real datasets. To achieve this, we switch from ModelNet10 to a subset of ModelNet40 dataset for the synthetic training data and additionally generate a custom test dataset by capturing 3D data using a depth camera. The switch is needed in order to be able to have corresponding objects in both the synthetic and training dataset. As a result, this does not allow a direct comparison of the results with other existing methods because we use different datasets. In addition, other methods do not consider partial 3D shapes and thus do not include any additional preprocessing to allow us to use synthetic data for training. However, by first evaluating our pipeline on ModelNet10, then switching to ModelNet40 and finally evaluating the combination of synthetic and real data, we try to ensure comparability for others.

## 3. Methodology

The processing steps for the training and classification data flows are summarized in Figure 1. The training data flow starts by performing different data preprocessing options for each synthetic object (shown as red blocks). For each object, we compute the 3DHOG object descriptor to generate a 3DHOG feature matrix. Then, we compute the principal components (PCs) of the feature matrix and the training data are projected onto a reduced subset of PCs. As a result, the feature matrix has a smaller dimensionality while maintaining most of the data variance. Finally, an object classifier is trained to estimate the object classes. The classification flow starts by measuring the real objects using a depth camera. Objects are segmented from the point cloud data by removing the GP and a clustering processing step. Additionally, segmented objects require preprocessing according to the one performed in the training flow. From each object, the 3DHOG object descriptor is computed to generate the feature vector. Then, each feature vector is projected onto the same reduced subset of PCs used during the training process and the object’s class is estimated using the same previous trained classifier.

### 3.1. Training Dataset and Data Preprocessing

We use a subset of the synthetic Princeton ModelNet40 dataset [7] to train the object recognition chain. It contains 3D volumetric images of 40 different object classes with separate sets for training and testing. However, we limited the evaluation to 10 classes choosing those classes where it is possible to capture enough real data. The selected object classes and the number of objects per class ($N_{Objects}$) are listed in Table 1. Due to the differences between the synthetic data used for training and the 3D data captured by the depth camera, we experimented with 5 different preprocessing options for the training data in order to determine the limitations when using a synthetic dataset for training. The preprocessing options are summarized as:No preprocessing. The object descriptor is computed directly for each object from the training dataset.Subset of training objects. We selected a subset of synthetic objects from the training dataset in order to train the recognition data flow. We evaluated all the training dataset objects at $20^{3}$, $30^{3}$ and $40^{3}$ voxel grid resolutions in order to discard those not having enough resolution. In order to determine the objects with a too low resolution, we manually evaluated each object one-by-one. The main criteria for the selection were that the object shows qualitatively distinctive features from the class it belongs to. Consequently, the evaluation contained all the objects at different voxel grid resolutions.Dataset augmentation. We perform a dataset augmentation by rotating each synthetic object from the training dataset along the Z axis. Hence, we generate multiple views of the same object as in Figure 2. We assume therefore, that the objects are always aligned on the X and Y axes. Otherwise, additional rotations on the X and Y axes are required.Frontal projection. Due to the lack of a complete 3D object shape captured by a depth camera, synthetic objects are preprocessed to compute the frontal projection according to the camera position described in Figure 3.Pose alignment. Each synthetic object from the training dataset is aligned using the PCA-STD method described in Vilar et al. [30] before computing the object descriptor in order to achieve rotational invariance.

### 3.2. Preparation of the Real Dataset

Our experimental object acquisition setup is shown in Figure 4. The depth camera is placed on a tripod tilted down 20 degrees. The objects are placed on a flat surface or ground plane (GP). Camera distance with respect to the objects is adjusted according with the size of the objects. As a result of the lack of a complete object shape measurement, it is required to perform a dataset augmentation by physically rotating the objects along the Z axis, Figure 5.

Other axis rotations are not considered for this experiment. The camera data are preprocessed in order to segment the objects from the point cloud data. All preprocessing stesp are shown in Figure 1 and summarized as follows:Point cloud conversion. The depth map image captured by the camera (Figure 4) is converted to a cloud of non-structured points in 3D space. The conversion is performed using the Application Programming Interface (API) software functions provided by the camera manufacturer.Ground plane removal. As a first object segmentation step, the point cloud points belonging to the GP are segmented and removed using the M-estimator sample-consensus (MSAC) algorithm [31]. In addition, the estimated GP points are used to align the remaining point cloud data with respect to the GP. This alignment is performed first by computing the principal component analysis (PCA) of the estimated GP points and later by performing an affine transformation of the remaining point cloud data in order to rotate it according with the measured angles (φ, α) (Equation 1) of the GP. Angle measurements are performed considering that the principal components on the Z axis ($\vec{PC3}$) are equivalent to the normal vector ($\vec{N}$) of the GP, Figure 6.
(1)$$\begin{array}{c} \alpha =-\arcsin\left|PC3(y)\right|\;\;\varphi =\arccos\left|PC3(x)\right| \end{array}$$Clustering. Point cloud data above the GP are segmented into clusters based on the Euclidean distance. Minimum Euclidean distance parameter is chosen according to the experimental results.Pose alignment. In the same way as described for the training dataset, the segmented objects are preprocessed to normalize their pose using the PCA-STD method described in [32] in order to achieve rotational invariance.Voxelization. Point cloud data from the segmented cluster are then converted into a voxel-based representation. This conversion is performed first by normalizing the data size according with the voxel resolution grid, and later by approximating each point form the point cloud to the nearest voxel integer value.

### 3.3. Object Descriptor and Dimensionality Feature Reduction

We used a handcrafted object descriptor in order to extract the key features from the objects regarding their respective class. The descriptor is based on a 3D voxel-based extension of the histogram-of-oriented-gradients (3DVHOG) [27] and was developed originally to detect hazard situations due to the presence of dangerous objects in a 3D scene. The descriptor parameters and feature dimensionality for each voxel grid analyzed are summarized in Table 2 and Table 3 and can be computed as is shown in Vilar et al [32].

According to the descriptor parameters shown in Table 3, $N_{Features}$ has a high dimensionality, demanding therefore higher computational and memory resources. In order to solve this limitation, we propose to reduce the feature dimensionality by computing the principal component analysis (PCA) of the feature matrix and select a reduced subset of PCs which contains most of the initial data variance. However, this dimensionality reduction method is especially difficult when dataset augmentation is performed. Additional data leads to an extremely high dimensionality feature matrix, demanding therefore a considerable additional training time.

### 3.4. SVM Classifier

In our previous work [8], we choose a multiclass support vector machine (SVM) classifier in order to learn the object classes from the feature matrix. The goal of that study was to compare the recognition performances with respect to other related approaches. To chose a suitable classifier, we evaluated different classifiers and different configuration parameters in that work and use the best option identified there for this study. Data parameters and configuration of the SVM classifier are shown in Table 4.

## 4. Results and Analysis

### 4.1. Experimental Flow

In order to evaluate the effect of the depth camera non-idealities and the different preprocessing and dataset options defined in Section 3.1 and Section 3.2, we defined the next sequence of experiments, Table 5.

### 4.2. Acquisition of Real World Data

We choose the active stereo-camera Intel RealSense D435: Intel, Santa Clara, CA, USA as a depth camera, Figure 7. Depth is measured directly in the camera by computing the pixel disparity using a variant of the semi-global matching algorithm on a custom ASIC processor. Hence, it is not required to include an additional processing task to measure the depth. The camera has a non-visible static infrared (IR) pattern projector to allow measuring the depth at dark-light conditions and also when the scene’s texture is too low. The Intel RealSense D435 camera uses a global shutter enabling robotic navigation and object recognition applications on a moving environment [30]. It has also a small size, making it suitable to be embedded into a robot or vehicle’s frame easily. In addition, the Intel RealSense D435 camera also integrates an RGB camera. However, we limited our research to depth images. Main camera specifications are summarized in Table 6.

### 4.3. Experiment 1

In experiment 1, we use a synthetic dataset for both training and classification data flows, without including any additional data preprocessing, Figure 1. The experiment goal was to verify previous recognition results using the ModelNet10 [7] synthetic dataset [8] and also determine the maximum ideal recognition accuracy defined as the averaged class accuracy ($ACC_{Class}$):(2)$$ACC_{Class}=\frac{1}{N}\sum_{i=1}^{N}\frac{\left(T_{p_{C}}+T_{n_{C}}\right)}{\left(T_{p_{C}}+T_{n_{C}}+F_{p_{C}}+F_{n_{C}}\right)}$$
where $T_{p_{C}}$ are the class C true positives, $T_{n_{C}}$ the class *C* true negatives, $F_{p_{C}}$ the class *C* false positives, $F_{n_{C}}$ the class *C* false negatives and *N* the number of classes.

Recognition accuracy for $20^{3}$,$30^{3}$,$40^{3}$ voxel resolutions and average of 3 different measurements are shown in Figure 8. Recognition results agree with previous research. The maximum $ACC_{Class}$ is 91.5% using 100 PCs and a $40^{3}$ voxel grid. In addition, the differences in $ACC_{Class}$ between the different voxel grids analyzed are small.

### 4.4. Experiment 2

In experiment 2, we use a synthetic dataset for both training and classification data flows and additionally included the PCA-STD pose-normalization preprocessing [32] to achieve rotation invariance. The experimental goal was to verify its performance using a new synthetic dataset with respect previous research. Experimental results show (Figure 9) equivalent results to the experiment 1, achieving a maximum $ACC_{Class}$ of 91% using 100 PC.

### 4.5. Experiment 3

In experiment 3, we use a synthetic dataset for training and the real dataset as test dataset. This dataset contains captures of real objects using the Intel RealSense D435 depth camera to evaluate the recognition performances. This experiment does not include any additional data preprocessing for both datasets. The experimental goal was to evaluate the recognition performance using real data from the depth camera without the requirement of generating a training dataset with real data. Our results show a maximum $ACC_{Class}$ of 68% using a voxel grid of $30^{3}$ voxels, Figure 10. In addition, $ACC_{Class}$ results without performing PCA are lower than in previous experiments, especially for the lowest voxel grid analyzed ($20^{3}$ voxels).

### 4.6. Experiment 4

In experiment 4, we use the same datasets as in experiment 3 for training and testing. However, the experiment includes the PCA-STD pose-normalization preprocessing [32] in order to achieve rotation invariance. The experimental goal was to verify the classification performance using real objects captured by the depth camera. Experimental results show a maximum $ACC_{Class}$ of 20% using 100 PCs without differences between the voxels grids analyzed, Figure 11.

### 4.7. Experiment 5

In order to improve the results of experiments 3 and 4 when using a real dataset, we perform a training dataset augmentation by rotating each object around the Z axis, Figure 3. Experimental results show an improvement with respect to the previous experiments, achieving a maximum $ACC_{Class}$ of 75.7% using a $40^{3}$ voxel grid, Figure 12. However, $ACC_{Class}$ is lower for the other voxels grids analyzed, especially for the $20^{3}$ grid.

### 4.8. Experiment 6

In experiment 6, we evaluated the effect of not having a complete 3D object shape in the synthetic training dataset. Hence, we removed the voxels not belonging to the frontal projection for each object of the training dataset,  Figure 3. Experiment results show a maximum $ACC_{Class}$ of 75.7% using a voxel grid of $40^{3}$ voxels and 500 PCs, Figure 13. In addition, higher voxel grids and thus high resolution objects perform better. However, it is required a significant higher number of PCs with respected previous experiments to achieve the best $ACC_{Class}$.

### 4.9. Experiment 7

In our last experiment, we redo experiment 6, but we modified the training dataset in order to remove the objects not having enough resolution after the voxel format conversion. The removed objects require using higher voxel grids than $40^{3}$ voxels in order to capture enough details to identify their class and therefore they do not match with the object resolution of the real dataset. Experimental results show an improvement with respect to results from experiment 6, achieving a maximum $ACC_{Class}$ of 81.5% using a $40^{3}$ voxel grid, Figure 14. However, it is required to use 300 PCs and thus a higher number of PCs in comparison with experiments 1 and 2. Other lower voxel grids analyzed achieve lower $ACC_{Class}$s, especially for the $20^{3}$ grid. Additionally, we compute the confusion matrix for the best case analyzed result, Table 7. All the objects classes are relatively well classified except for the classes 8 (Monitor) and 10 (Stool).

## 5. Discussion

### 5.1. Synthetic Dataset

In experiments 1, Figure 8, and 2, Figure 9, we evaluated the recognition performance of the 3DVHOG using a subset of classes of the ModelNet40 dataset. In both experiments, recognition accuracy is up to 90% and thus comparable to the state-of-the-art 3D recognition approaches as well as our previous results [8]. The best results are achieved using the highest voxel grid ($40^{3}$), but the other voxel grids analyzed also achieved equivalent recognition results while reducing considerably the computational cost and memory requirements. However, a synthetic test dataset does not allow to evaluate the 3DVHOG descriptor in a real application due to the camera limitations and differences between synthetic and real objects.

### 5.2. Real Dataset

In experiment 3, Figure 10, we evaluated the recognition accuracy when a depth camera is used to capture 3D objects instead of using a synthetic test dataset. As expected, the results show lower recognition accuracy than previous experiments due to the differences between both synthetic and real objects. The main problem is the limitation to a partial object shape when using the real data due to occlusion. This and other differences require to include some additional data preprocessing. In addition, results show that it is required to use higher voxel grids in order to improve the recognition results when using a real dataset. Higher voxel grids capture more object details and therefore the 3DVHOG descriptor is able to capture enough information to allow the SVM classifier to estimate the object’s class. Hence, one solution to solve the problem of having partial object shapes when using the 3DVHOG descriptor is to increase the object´s resolution of the synthetic training dataset. This can be achieved by using higher voxel grids but also by increasing the level of detail captured by the 3DVHOG descriptor, i.e., by using higher resolutions for angle bins ($\varphi_{Bins}$, $\theta_{Bins}$). This solution is therefore equivalent to use a local descriptor instead of a global one. However, the computational cost and memory requirements are increased cubically making it, as consequence, not suitable for a real object recognition application.

### 5.3. Rotational Invariance Data Preprocessing

The first data preprocessing analyzed is the PCA-STD method in order to achieve rotational invariance for the 3DVHOG descriptor, Figure 9 and Figure 11. When it is used with a synthetic test dataset, PCA-STD performs correctly, but when a real test dataset is used, the method fails. The reason for these bad results is that the real objects have only partial shapes in comparison to the synthetic ones due to the limitations of the camera measurement. Thus, object shapes and object geometry are different in both datasets and thus the standard data deviation for each class. Consequently, it is required to explore alternative methods for rotational invariance when dealing with partial object shapes and thus real data.

One method for rotational invariance, explored in experiment 5, Figure 12, is to perform a training dataset augmentation by rotating each object along the Z (vertical) axis to generate additional objects which contain most of the data variance, Figure 8. Results show a significant recognition accuracy increment with respect to experiment 4. However, they are not comparable with those in experiments 1 and 2, when a synthetic test dataset is used. In addition, a lower voxel grid performs worse in all cases analyzed. Despite the partial object shapes, it is required to increase the level of object´s detail in order to capture local object features regarding their class. In addition, a dataset augmentation leads to increase exponentially the size of the training dataset and therefore to extend exponentially the required time for training the classifier.

### 5.4. Frontal Projection Data Preprocessing

In experiment 6, Figure 13, we added another preprocessing step for the synthetic objects of the training dataset to compute the frontal projection in order to increase the recognition accuracy when using a real test dataset. However, results show a relatively small recognition accuracy improvement with respect to experiment 5, Table 5. In addition, voxel grids of $30^{3}$ and $40^{3}$ achieve the same recognition results contradicting our initial results, that higher resolutions are required to capture enough details. We then analyzed the objects in the synthetic training dataset to check for potential problems and differences in the data used for training and test.

### 5.5. Subset of Synthetic Training Objects

Finally, in experiment 7, Figure 14, we continue exploring the idea to adapt as much as possible the training dataset to the real dataset by selecting the synthetic training objects according with those used in the real dataset, Table 1. Results show an increment of the recognition accuracy, but are still not comparable with the results achieved in idealistic experiments 1 and 2. Confusion matrix (Table 7) for the best case analyzed shows relatively well classified objects, except for the classes 8 (Monitor) and especially for 10 (Stool). After reviewing some the objects of classes 8 and 10, Figure 15, it is possible to identity the next error sources:Glossy and no-reflective surfaces. Monitors have some areas where the structured IR light pattern is not reflected back to the camera. This error depends on the relative angle between the monitor and the camera. Thus, at certain angles, the camera can not capture the object and therefore compute any depth. As a result, it appears as a blank area in the depth computation, causing a complete distortion of the original object shape. In addition, the same effect occurs on glossy surface areas where the IR light pattern of the camera is reflected back in multiple directions.This camera drawback is specially relevant for robotic applications where it is required to detect and track surrounding objects or humans. Light absorbent materials (e.g., dark clothes) and light reflective materials can distort the depth measurement and thus limit their application scope when camera uses active illumination [30].Segmentation errors. Object segmentation relies on the GP’s flatness and the MSAC algorithm to estimate the GP points. Noise and depth computation errors can lead to an incomplete removal of the GP from the point cloud data. As a consequence, objects in the scene are not segmented correctly causing a distorsion of the original object shapes.Depth artifacts. Sharp and narrow edges in the camera image lead to shadow errors in the 3D point cloud generation and thus distort completely the object shape measurement.

### 5.6. Response Time

Although 3DHOG response time is out of the scope of this paper, we consider it relevant to discuss the timing constraints for a real-time implementation. We measured the response times of experiment 2, using a synthetic dataset, and experiment 7, using a real dataset, for the best results in terms of recognition accuracy and required number of PCs, Table 8. Our results show that when a real dataset is used, the significantly higher number of PCs (300) to achieve equivalent recognition accuracy results increases, as a consequence, the response time. The response time is increased because the classifier requires more time ($t_{Class}=180$ ms) to process the higher dimensionality of real data with respect the synthetic dataset ($t_{Class}=38$ ms). The other measured response times, the pose normalization time ($t_{N}$), the 3DHOG computation time ($t_{3DHOG}$) and the projection time onto the PC ($t_{PC}$) are shorter or equivalent in all cases. As a consequence, the main limitations to achieve a real-time performance are the 3DHOG dimensionality and the number of required PCs to improve the recognition accuracy. The goal should be to reduce the feature dimensionality while maintaining the recognition accuracy at the same time. We believe that, due to lack of completed 3D shape, the intra-class object variability is higher in comparison with the synthetic dataset, requiring a higher number of PCs and thus higher response times.

For real-time applications, this requires a careful balancing between accuracy and dimensonality or the evaluation of alternative approaches to recognize the objects. This is even more important since the measured times will increase further, since the current evaluation did not include the image acquisition and object segmentation required for the real data pipeline.

Custom test datasets do not allow a side-by-side comparison of different approaches, especially with small datasets. We instead summarize some of the approach specifications and results in Table 9.

The 3DHOG results show a recognition accuracy slightly lower than the state-of-the art CNN object recognition approaches but still in line with them. Regarding the timing, our approach shows higher values than the CNN-based approaches. This results from the required increased feature dimensionality to handle real data and corresponds to the results in Simon et al. [12] that uses Lidar data. Other global approaches show a similar performance, even if training and test datasets consist of real data. The 3DHOG response time is lower than the local approaches. These approaches require around 10 s per classification and are thus infeasible for real-time applications.

In addition, our approach is currently not optimized for execution performance, since we use a single thread in order to estimate single core embedded timing. This results in further optimization potential in order to reduce the response times and enhance our approach. At this stage, our approach shows that it is possible to use synthetic data for training a processing pipeline that is afterward dealing with real data while maintaining a reasonable classification accuracy and response time.

## 6. Conclusions

In this paper, we aimed to analyze which steps are needed to transfer an existing 3DHOG-based recognition approach from synthetic data to work with real data. To address the challenge of a limited dataset during the design phase of the system, we propose to continue using synthetic data for training and real data for testing the classifier. Synthetic data have an advantage for the training as it is easier to generate an appropriate number of samples from complete object models. Due to the differences between synthetic data and real data, the proposed preprocessing data flow needs several adjustments.

Our experimental works show that the 3DVHOG performs correctly when real data are used for the test and a synthetic dataset for training, showing that this approach is applicable to other 3D recognition tasks as well. However, it is required to include additional data preprocessing steps on the training dataset as well as adapted processing steps for the real data in order to maximize the recognition accuracy. The PCA-STD method for 3DVHOG rotational invariance is not suitable when evaluating with real data due to the object differences between both datasets. Instead, it is required to perform a training dataset augmentation to achieve rotation invariance. The evaluated preprocessing data flow improves the recognition performances, but requires to use higher resolution of the voxel grid and a significantly high number of PCs compared to an ideal case with synthetic data only. Despite that, our recognition results show that most of the classes are well classified except for the classes “Stool” and “Monitor” due to segmentation errors and camera measurements artifacts.

In order to further improve the recognition results, we propose to reduce the intra-class data variability of the synthetic training dataset, especially for the classes “8—Monitor” and “10—Stool” and include additional preprocessing to compensate the non-idealities in the camera measurements and thus match the synthetic and real objects datasets.

## Figures and Tables

**Figure 1 sensors-21-00910-f001:**
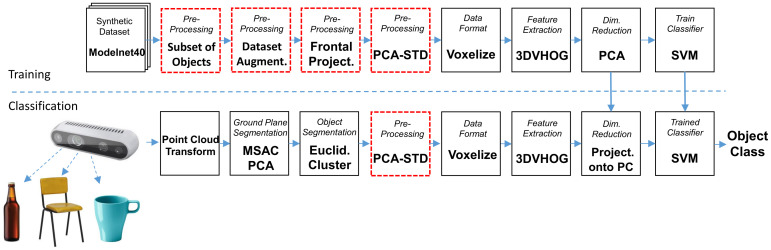
Training and classification data flows.

**Figure 2 sensors-21-00910-f002:**
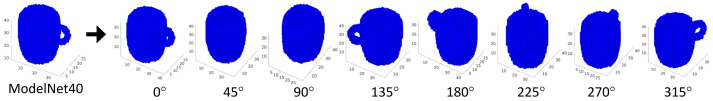
Training dataset augmentation by rotating each object along the Z axis 0°, 45°, 90°, 135°, 180°, 225°, 270°, 315°.

**Figure 3 sensors-21-00910-f003:**
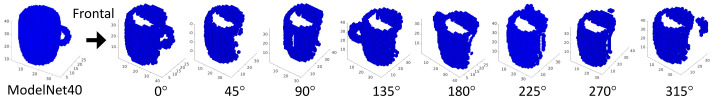
Training dataset augmentation by rotating each object along the Z axis 0°, 45°, 90°, 135°, 180°, 225°, 270°, 315° including only the frontal projection of the object with respect to the camera position.

**Figure 4 sensors-21-00910-f004:**
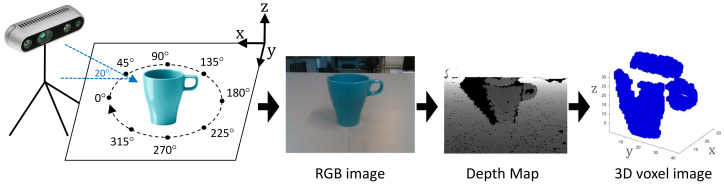
Evaluation setup, RGB image, depth map and object representation in voxels.

**Figure 5 sensors-21-00910-f005:**
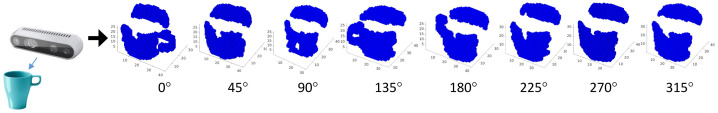
Real dataset augmentation by rotation of each object along the Z axis 0°, 45°, 90°, 135°, 180°, 225°, 270°, 315°.

**Figure 6 sensors-21-00910-f006:**
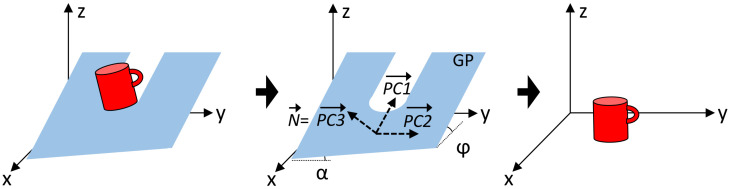
Pointcloud data alignment by computing the principal component analysis (PCA) of the estimated ground plane (GP).

**Figure 7 sensors-21-00910-f007:**
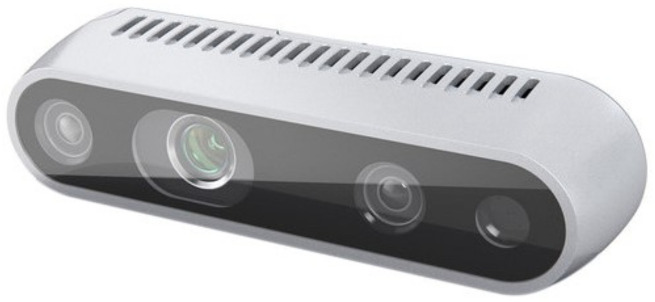
Intel RealSense D435 depth camera.

**Figure 8 sensors-21-00910-f008:**
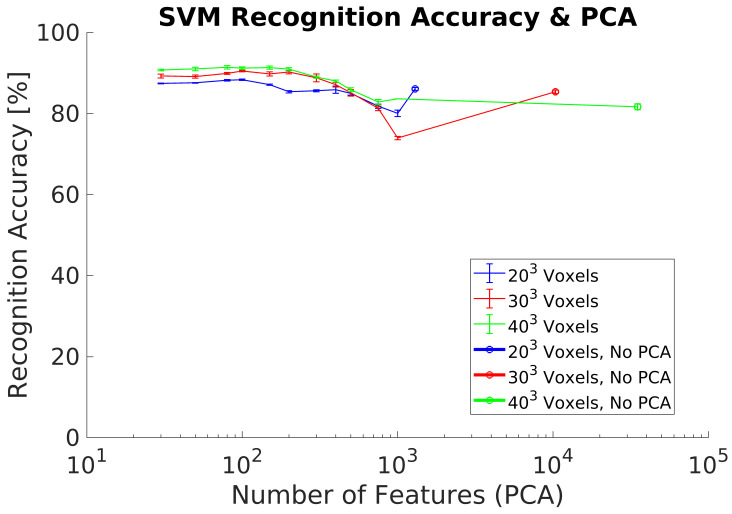
Experiment 1. Recognition accuracy without including preprocessing using a synthetic dataset.

**Figure 9 sensors-21-00910-f009:**
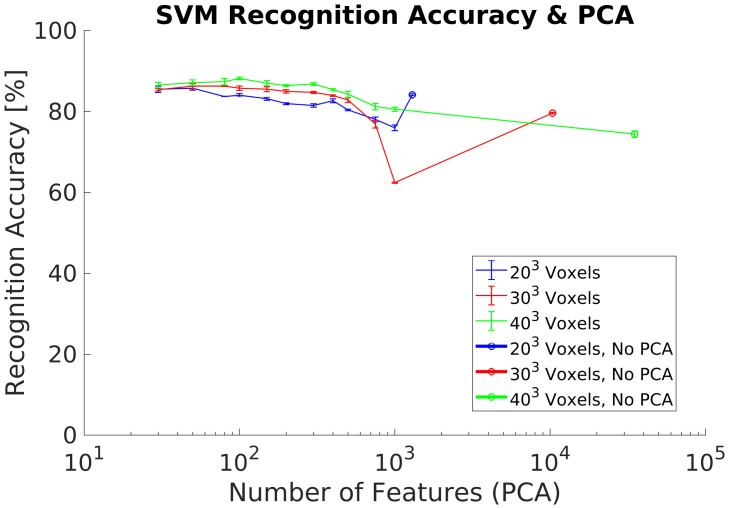
Experiment 2. Recognition accuracy including pose normalization using a synthetic dataset.

**Figure 10 sensors-21-00910-f010:**
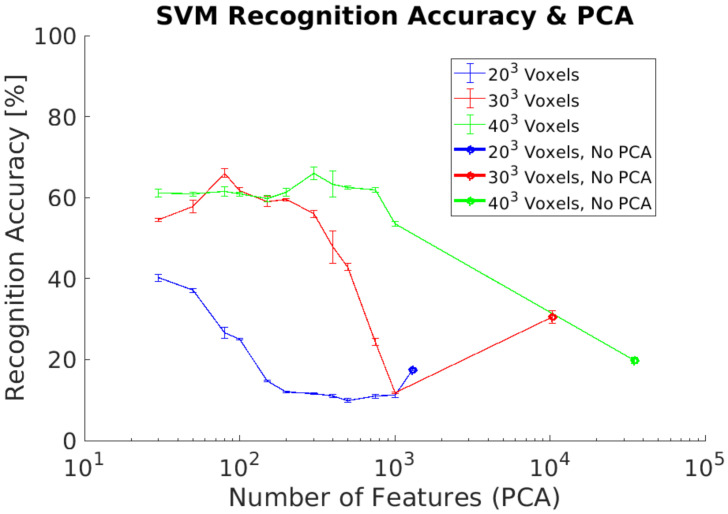
Experiment 3. Average recognition accuracy without including preprocessing using a real dataset for classification.

**Figure 11 sensors-21-00910-f011:**
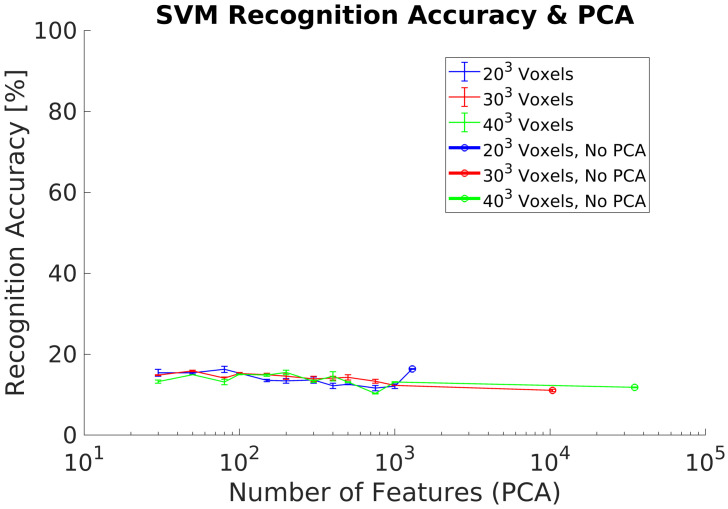
Experiment 4. Average recognition accuracy including pose normalization using a real dataset for classification.

**Figure 12 sensors-21-00910-f012:**
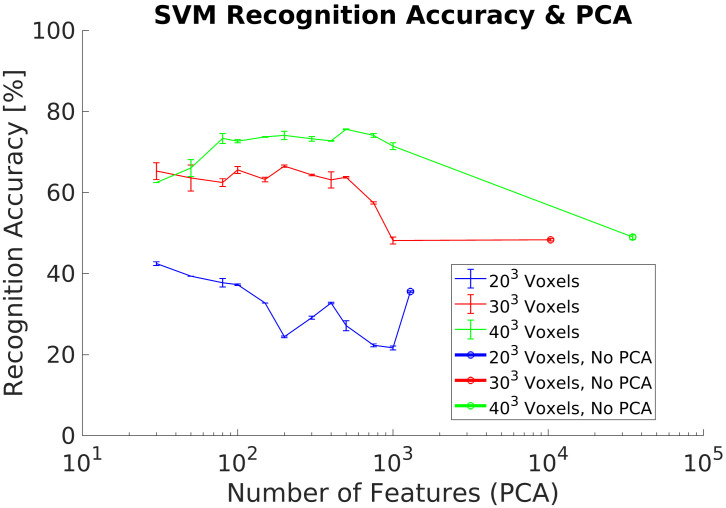
Experiment 5. Average recognition accuracy on a real dataset using a synthetic training dataset including dataset augmentation preprocessing.

**Figure 13 sensors-21-00910-f013:**
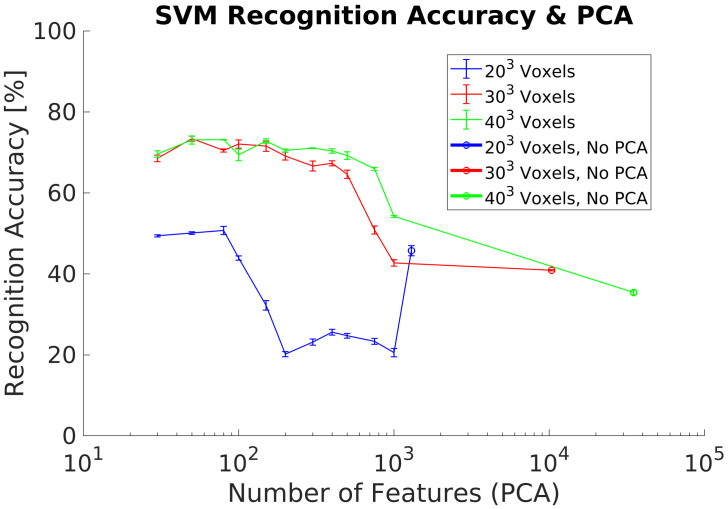
Experiment 6. Average recognition accuracy on a real dataset using a synthetic training dataset including frontal projection and dataset augmentation preprocessing.

**Figure 14 sensors-21-00910-f014:**
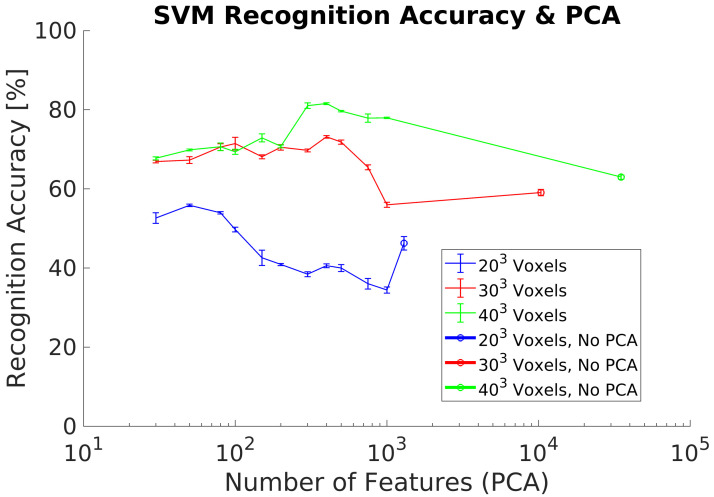
Experiment 7. Average recognition accuracy results including dataset augmentation, frontal projection and a subset of synthetic training objects.

**Figure 15 sensors-21-00910-f015:**
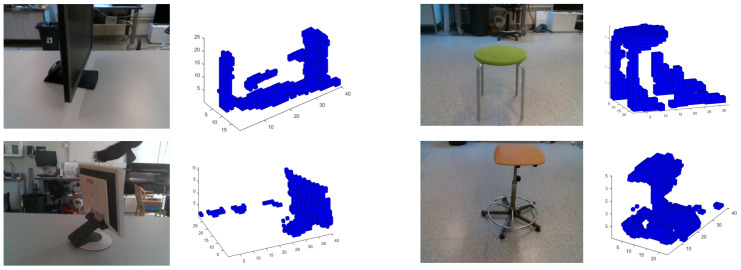
Example of objects in class 8 (Monitor) and class 10 (Stool) captured by the Intel RealSense D435.

**Table 1 sensors-21-00910-t001:** ModelNet40 object classes selected and number of objects per class for the synthetic and real datasets.

Dataset	Bottle	Bowl	Chair	Cup	Keyboard	Lamp	Laptop	Monitor	Plant	Stool	$N_{\mathrm{Objects}}$
Training	335	64	889	79	145	124	149	465	240	90	2580
Augmentation.
Training	2680	512	7112	632	1160	992	1192	3720	1920	720	20,640
Sub.&Augmentation.
Training	816	448	1096	288	560	472	536	712	648	528	6104
Synthetic
Test	99	19	99	19	19	19	19	99	99	19	510
Camera
Test	64	48	64	64	48	48	48	48	48	48	528

**Table 2 sensors-21-00910-t002:** 3D histogram-of-oriented-gradients (3DHOG) and feature dimensionality for $20^{3}$, $30^{3}$, $40^{3}$ voxel grids.

Voxel Grid	$20^{3}$	$30^{3}$	$40^{3}$
$N_{Blocks}$	1	8	27
$N_{Cells}$	8	8	8
$N_{Features}$	1296	10,368	34,992

**Table 3 sensors-21-00910-t003:** Descriptor configuration parameters.

Parameter	Value
$\varphi_{Bins}$	18
$\theta_{Bins}$	9
$Cell_{Size}$	6
$Block_{Size}$	2
$Step_{Size}$	2

**Table 4 sensors-21-00910-t004:** Support vector machine (SVM) classifier configuration parameters.

Parameter	SVM Classifier
Type	Multiclass
Method	Error correcting codes (ECOC)
Kernel function	Radial basis function
Optimization	Iterative single data (ISDA)
Data division	Holdout partition 15%

**Table 5 sensors-21-00910-t005:** Experiments definition (Exp.), experimental flow and maximum recognition accuracy (Acc) achieved.

Exp.	Training Dataset		Test Dataset		Acc. Results
1	Synthetic	No preprocessing	Synthetic	No preprocessing	91.5%
2	Synthetic	PCA-STD alignment	Synthetic	PCA-STD align.	90%
3	Synthetic	No preprocessing	Real	No preprocessing	65%
4	Synthetic	PCA-STD alignment	Real	PCA-STD align.	21%
5	Synthetic	Dataset augm.	Real	No preprocessing	75.7%
6	Synthetic	Dataset augm. + Frontal view	Real	No preprocessing	73.7%
7	Synthetic	Dataset augm. + Frontal view + Subset	Real	No preprocessing	81.5%

**Table 6 sensors-21-00910-t006:** Intel RealSense D435 active stereo-camera specifications.

Technology	Active IR Stereo
Sensor Technology	Global shutter, 3 × 3 μm
Depth Field-of-View	86${}^{\circ }$ × 57${}^{\circ }$
Depth Resolution	up to 1280 × 720 pixels
RGB Resolution	1920 × 1080 pixels
Depth Frame Rate	up to 90 fps
RGB Frame Rate	30 fps
Min Depth range	0.1 m
Max Depth range	10 m
Dimensions	90, 25, 25 mm

**Table 7 sensors-21-00910-t007:** Normalized confusion matrix using 300 principal components (PCs) and synthetic training dataset and a $40^{3}$ voxel grid with frontal view and a subset of objects.

Estimated Class	1	0.98	0	0	0.04	0.02	0	0.02	0.02	0	0
	2	0	0.98	0.03	0	0	0	0	0	0	0
	3	0	0	0.86	0	0	0	0	0	0.02	0.2
	4	0.01	0	0	0.86	0	0.14	0	0	0	0
	5	0	0	0	0	0.95	0	0	0	0	0.02
	6	0	0	0.01	0.04	0	0.85	0.04	0.12	0	0.04
	7	0	0.02	0	0	0.03	0	0.73	0.18	0	0.04
	8	0	0	0.09	0.05	0	0	0.15	0.60	0	0.40
	9	0.01	0	0.01	0.01	0	0.01	0.06	0.06	0.85	0
	10	0	0	0	0	0	0	0	0	0.13	0.30
		1	2	3	4	5	6	7	8	9	10
		Input Class									

**Table 8 sensors-21-00910-t008:** Response times for a $40^{3}$ voxel grid (1) using the synthetic dataset and 100 PCs, (2) using a real dataset and 300 PCs.

Test Dataset	$N_{\mathrm{Features}}$	PC	Acc (%)	$t_{N}$ (μs)	$t_{3\mathrm{DHOG}}$ (*ms*)	$t_{\mathrm{PC}}$ (*ms*)	$t_{\mathrm{Class}}$ (*ms*)	$t_{\mathrm{Total}}$ (*ms*)
Synthetic	34,992	100	90	9	11.5	6	20.5	**38**
Real	34,992	300	81.5	—	12	4	180	**196**

**Table 9 sensors-21-00910-t009:** Summary of the 3DHOG descriptor and results comparison with respect to other 3D object recognition approaches.

Approach	Method	Training Dataset	Test Dataset	3D Sensor	Accuracy	GPU	$T_{\mathrm{Total}}$
**3DHOG**	**Global**	**ModelNet40**	**Real Custom**	**Stereo**	**81.5%**	**No**	**196 ms**
**3DHOG** [32]	**Global**	**ModelNet10**	**ModelNet10**	–	**84.91%**	**No**	**21.6 ms**
VoxNet [11]	CNN	ModelNet10	ModelNet10	–	92%	Yes	3 ms
PointNet [10]	CNN	ModelNet10	ModelNet10	–	77.6%	Yes	24.6 ms
3DYolo [12]	CNN	KITTY	KITTY	Lidar	75.7%	Yes	100 ms
SPNet [13]	CNN	ModelNet10	ModelNet10	–	97.25%	Yes	–
VFH [23]	Global	Real Custom	Real Custom	Stereo	98.52%	–	–
SI [22,23]	Global	Real Custom	Real Custom	Stereo	75.3%	–	–
VFD [6]	Global	ModelNet10	ModelNet10	–	92.84%	No	–
RCS [17]	Local	UWA	UWA	–	97.3%	No	10–40 s
TriLCI [18]	Local	BL	BL	–	97.2%	No	10 s

## Data Availability

The data presented in this study are available on request from the corresponding author.

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
