# Peer review of "Realworld 3D Object Recognition Using a 3D Extension of the HOG Descriptor and a Depth Camera"

_sensors, 2021, doi:10.3390/s21030910_

Round 1

Reviewer 1 Report

The work is well structured and easy to read. The main contribution of the paper is a study on 3D object recognition, which uses real data from a depth camera in c classification and synthetic data to train the classifier. To evaluate recognition performance, experiments are performed for various combinations of test and training data, as well as different pre-processing steps in the classification process.

The reported results are acceptable. In general I find the proposal very interesting and with great applicability.

Minor suggestions:

  1. For a better understanding of the paper, organize the explanation of the preprocessing options in the same order that they appear in figure 1.
  2. Line 210 - Figure3
  3. Line 218- 403
  4. Use the same format: Experiments x or experiment x, Figure x or figure x
  5. Use the same format in references.

Author Response

Reviewer 1

The work is well structured and easy to read. The main contribution of the paper is a study on 3D object recognition, which uses real data from a depth camera in c classification and synthetic data to train the classifier. To evaluate recognition performance, experiments are performed for various combinations of test and training data, as well as different pre-processing steps in the classification process.

The reported results are acceptable. In general I find the proposal very interesting and with great applicability.

Minor suggestions:

  • For a better understanding of the paper, organize the explanation of the preprocessing options in the same order that they appear in figure 1.

The explanation of the preprocessing options has been reorganized according toFigure 1 in Section 3. In addition, we included an extended explanation for Figure 1 at the beginning of Section 3.

  • Line 210 – Figure3

Fixed.

  • Line 218- 403

Fixed

  • Use the same format: Experiments x or experiment x, Figure x or figure x

Fixed. We use always “experiments” and “Figure”.

  • Use the same format in references.

Fixed

Reviewer 2 Report

The paper presents an interesting subject but the following aspects must be detailed:

  • explain more clearly Figure 1 in Section 3
  • explain more clearly: line 112 "discard those not having enough resolution." - how was chosen the threshold for this resolution?
  • line 114: "object acquisition setup is shown in Figure 2." - may be is another figure, but not figure 2
  • how was chosen parameters for the SVM (from Table 4)?
  • for Table 7 it was better to use a normalised confusion matrix (not every class has the same number of elements)
  • what is the response time - is it possible to use the method in real time?
  • the obtained results must be compared with other existing methods
  • place Table 1 after its reference in the text (the same for figures 8-14)

Round 2

Reviewer 2 Report

Some of my comments were addressed. There is one comment that is not clearly explained - regarding the comparison of the proposed method with other existing ones.

The response is : line 86: "Our previous 3DHOG descriptor results are compared with other methods in our previous publications [8] using the ModelNet10 synthetic dataset"

Since the results of the previous proposed method are already compared with other existing papers, there is a need to clarify: what are the differences between your previous paper and the current one. Since there is another paper there must exist some differences and in this case the current results must be compared with other existing methods even if your method is evaluated with your own dataset (there are methods for which the code is free on the internet and they can be tested with other datasets).

Round 3

Reviewer 2 Report

Since all my comments were addressed I recommend to publish the paper.